# QSPR/QSAR: State-of-Art, Weirdness, the Future

**DOI:** 10.3390/molecules25061292

**Published:** 2020-03-12

**Authors:** Andrey A. Toropov, Alla P. Toropova

**Affiliations:** Laboratory of Environmental Chemistry and Toxicology, Department of Environmental Health Science, Istituto di Ricerche Farmacologiche Mario Negri IRCCS, Via Mario Negri 2, 20156 Milano, Italy; andrey.toropov@marionegri.it

**Keywords:** QSAR evolution, multi-target QSAR, Monte Carlo method, fuzzy sets

## Abstract

Ability of quantitative structure–property/activity relationships (QSPRs/QSARs) to serve for epistemological processes in natural sciences is discussed. Some weirdness of QSPR/QSAR state-of-art is listed. There are some contradictions in the research results in this area. Sometimes, these should be classified as paradoxes or weirdness. These points are often ignored. Here, these are listed and briefly commented. In addition, hypotheses on the future evolution of the QSPR/QSAR theory and practice are suggested. In particular, the possibility of extending of the QSPR/QSAR problematic by searching for the “statistical similarity” of different endpoints is suggested and illustrated by an example for relatively “distanced each from other” endpoints, namely (i) mutagenicity, (ii) anticancer activity, and (iii) blood–brain barrier.

## 1. Introduction

Each science meets with internal and external contradictions. Correlations many times have served as a key to the interpretation of various phenomena. Expansion of information available for analysis (e.g., search space along “traditional” substances has been extended by nanomaterials) leads to the following question: is the correlation useful or it will better to try to define a causality? [1]. Apparently, an answer to this question is deemed to be non-completed. At the first stage, Wiener had established the first correlations for physicochemical endpoints [2,3,4]. Later, other authors [5,6,7,8,9] continued the stream of similar studies. At the second stage (after almost twenty years pause), Hansh and Fujita [10] established the first correlations for biochemical endpoints.

The quantitative structure–property activity relationships (QSPRs/QSARs) are a relatively new field of natural sciences. There is a large group of aims associated with the QSPRs/QSARs technique, the main ones of these are probably the following: (i) prediction of the physicochemical behavior of various substances in industry and their further ecologic impacts [2,3,4,5,6,7,8,9]; (ii) the biochemical behavior of various substances in ecological and medicinal aspects [10]; (iii) selection of substances, which can be prospective candidates to the defined role [11,12].

Results of traditional experiments depended on properties of substances, masses, radiation; heat capacity, electronic, physicochemical, and biochemical conditions as well as porosity, Zeta potential of nanomaterials, time of exposure, irradiation, darkness, etc. Computational experiments related to QSPR/QSAR concerned with “information conditions” (available datasets) and “statistical conditions” (diversity of substances in datasets), as well as preference of the user.

Wiener has carried out the pioneer works in the field of correlation “molecular structure–macro-effect of a substance” in the 1940s [2,3,4]. This was the start of QSPR/QSAR history. In other words, this is the first stage of evolution of QSPR/QSAR theory and practice.

The main task of the QSPR/QSAR at this period was to establish a correlation between an endpoint and descriptor for a set of substances. Criteria of quality of those models were (i) the total number of compounds in the available set; (ii) correlation coefficient; (iii) standard error of estimation; and (iv) the Fischer F-ratio [1,2,3,4,5]. In this period the family of topological indices [6,7,8,9,10,11,12,13,14,15,16,17], indices based on the mathematical theory of information [18,19,20,21,22,23,24,25,26,27,28], various 3D descriptors [29,30,31,32], and descriptors of quantum mechanics [33,34,35,36,37,38] were the basis of the QSPR/QSAR theory and practice.

However, the absence of the reliable statistical checking up of these models had led to intense criticism of the QSPR/QSAR research. This criticism is continuing up to now [39,40,41,42].

A set of principles was proposed for evaluating the validity of QSAR models at a conference held in Setubal, Portugal in 2002. According to the Setubal principles, QSARs should:Be associated with a defined endpoint of regulatory importance;Take the form of an unambiguous algorithm;Have a clear domain of applicability;Be associated with appropriate measures of goodness of robustness, and predictivity;Have a mechanistic interpretation.

In further, these principles were renamed in OECD principles (Organization for Economic Co-operation and Development) http://www.oecd.org/chemicalsafety/risk-assessment/37849783.pdf

The OECD principles open the second stage of QSPR/QSAR history: “not only to establish a correlation, but to check up the predictive potential of the correlation.”

## 2. QSPR/QSAR: State-of-Art

There has been an improvement in the QSPR/QSAR technique during the last decade. However, some “unpleasant peculiarities” remain. The list of “main unpleasant peculiarities” of QSPR/QSAR analysis is as follows: (i) possibility of “chance correlations” [43,44,45,46]; (ii) possibility of overtraining [47]; (iii) possibility of weak reproducibility of statistical quality of an approach suggested [48,49].

A person who would like to apply a model hardly will be pleasant to the necessity to get a group of descriptors via hard-to-understand software and with the further necessity to carry out calculations with other hard-to-understand software that provides multiple linear regression analysis or the artificial neuron networks or something else. Attempts to solve problems related to the above “unpleasant peculiarities” of QSPR/QSAR have been performed. However, these attempts gave three weirdness points.

### 2.1. The First Weirdness of QSPR/QSAR 

The distribution of available data for QSPR/QSAR analyses into the training and validation sets can be done by various manners [50,51]. The distribution has a key influence on the statistical quality of QSPR/QSAR models [52,53]. Here, one can see first weirdness in the modern QSPR/QSAR researches: the majority of the models are based solely on distribution available data into the training and validation sets.

According to many authors, a rational split into training and validation set gives better statistical results of the validation sets than models based on random splits [54]. However, the experiment confirms that there are splits successful for one approach, which are unsuccessful for another approach [55,56,57,58,59]. For example, three different splits (Table 1) into training and validation sets of 87 anticancer inhibitors [60] give models with different predictive abilities (Table 2).

An examination of several splits decreases the probability of “chance correlations”: solely one good correlation easily can become chance correlation, however, three (five, six, seven, ...) good correlations hardly can be “chance correlations”.

Method 1 is a one-variable model calculated with the Monte Carlo technique [61,62,63,64] for hybrid optimal descriptors, which are calculated by simplified molecular input-line entry system (SMILES) [65], together with a molecular graph [66,67,68,69,70,71]:(1)$$DCW\left(1,10\right)=\sum CW\left(EC0_{k}\right)+\sum CW\left(EC1_{k}\right)+\sum CW\left(S_{j}\right)+\sum CW\left(SS_{j}\right)$$

The *EC0_k_* is the vertex degree in hydrogen-suppressed graph (HSG); *EC1_k_* is Morgan extended connectivity [72,73] of the first order; *S_j_* is SMILES atoms, i.e., one symbol (e.g., ‘C’, ‘N’, ‘O’, etc.) or a group of symbols which cannot be examined separately (e.g., ‘Cl’, ‘Br’, ‘@@’, %12, etc.); the *SS_j_* are connected pairs of the SMILES atoms.

Method 2 is a one-variable model calculated with the Monte Carlo method for hybrid descriptors:(2)$$DCW\left(1,10\right)=CW\left(C5\right)+CW\left(C6\right)+\sum CW\left(S_{j}\right)+\sum CW\left(SS_{j}\right)$$

The C5 and C6 are codes of molecular rings extracted from the adjacency matrix of HSG [74].

The *CW(EC0_k_), CW(EC1_k_), CW(S_j_), CW(SS_j_), CW(C5),* and *CW(C6)* are correlation weights of the above-listed SMILES attributes and invariants of HSG calculated with the Monte Carlo method (http://www.insilico.eu/coral).

The described experiment confirms successful and unsuccessful splits exist. Excellent split (Split 1) for the 3D-QSAR approach is poor for 2D approaches, i.e., models calculated by Equation (1) or Equation (2). However, (Table 2), Split 2 is excellent (at least successful) for Method 1, whereas the Split 3 is excellent (at least successful) for Method 2.

### 2.2. The Second Weirdness of QSPR/QSAR

The number of statistical characteristics aimed to measure the predictive potential of a model gradually increase (Table 3), despite the apparent attractiveness of a small number of criteria of the predictive potential for practical applications.

On the one hand, the diversity of different criteria of predicting potential is a tool to improve the quality of QSPR/QSAR models. On the other hand, this situation causes sometimes the uncertainty in the choice of the best model. In other words, contradictions in the recommendations of various criteria force the researcher to search for truth (i.e., the best choice) in a greater maze of possibilities.

### 2.3. The Third Weirdness of QSPR/QSAR

Naturally, the contribution of the molecular structure is the key importance to an endpoint. However, any biological activity is a mathematical function of many different conditions and circumstances. In other words, toxicity or pharmaceutical effect is caused by not only molecular structure, but also physicochemical conditions (e.g., temperature, humidity) and circumstances (noise/silence, illumination/darkness). Apparently, one can disagree with the above postulation, but the majority of QSPR/QSAR has built up without taking into account something besides molecular structure.

It is to be noted, however, in some cases, the molecular structure is not informative to build up a predictive model of endpoints [81,82,83,84,85,86,87,88,89,90,91,92,93,94,95]. Meanwhile, the definition of a model as a mathematical function of experimental conditions (after consultations with experimentalists) is a shorter and consequently more attractive way to solve the corresponding task.

## 3. Discussion

The above-mentioned unpleasant peculiarities and weirdness are interacting. To avoid unpleasant peculiarities, one should build up a model without the above weirdness, namely, (i) one should study several different splits (into training and validation sets); (ii) one should select a group of criteria of predictive potential which are agreed with each other; and (iii) one should take into account all conditions which impact corresponding endpoint (not only molecular structure). However, these actions are not enough to solve all problems.

Unfortunately, there are other problems. Fortunately, there are other solutions. The hierarchy of problems in the field of the modeling of various endpoints is not established. One group of researchers believes that the validation of a model is of key importance. Another group believes that the main result is the statistical quality of a model. A third group concentrates on mechanistic interpretation. It is curious, but non-standard tasks and solutions also exist and sometimes these are very important. Examples are below.

### 3.1. Multi-target QSAR Models

The limitation of almost all QSAR models is that they predict the biological activity for only one endpoint. In other words, traditional QSAR gives a model for the biological activity of drugs against only one parasite species [96], one species of a virus [97], and one type of cancer [98]. The so-called multi-target QSAR as a tool to build up models for several endpoints is suggested [96,97,98].

Apparently, this conception has attractive advantages, since it provides a user by extending a list of information (i.e., expected numerical data for groups of endpoints, which affect the phenomenon under consideration, e.g., therapeutic effect, inhibition, biocide potential, etc.). Nonetheless, traditional approaches serve as a basis to solve the task of building up multi-target QSARs, e.g., using multiple regression [99], partial least squares (PLS) [100], artificial neural networks (ANN) [101,102,103], and random forest [104].

It is to be noted, that interest to researches dedicated to multi-target QSAR in drug discovery gradually increases during the past decade, whereas interest to general QSAR in drug discovery is approximately constant. Figure 1 confirms this situation.

### 3.2. Similarity of Endpoints

As noted in the previous section, the simultaneous examination of two endpoints is an attractive way in the QSPR/QSAR analysis. In addition to multi-target QSAR, the similarity of endpoints may be a heuristic tool of control of the biochemical knowledge [105,106,107]. Similarity/dissimilarity of endpoints can be expressed via correlation weights of molecular features extracted from SMILES [105]. In principle, the spectrum of physicochemical conditions with a clear impact on biochemical endpoints (toxicity, therapeutic potential) able to provide hints to establish similarity (dissimilarity) for two endpoints relevant to drug discovery, toxicity, risk assessment, and others.

The similarity of endpoints was analyzed in the literature [105]. The task is to extract molecular features involved in the modeling process, which play an analogous role for corresponding models of (i) mutagenicity; (ii) blood–brain barrier; and (iii) anticancer activity.

#### 3.2.1. Mutagenicity

The endpoint for QSAR analysis is the mutagenic potential. The mutagenic potential in *Salmonella typhimurium* TA98+S9 microsomal reparation is represented by the natural logarithm of R, where R is the number of revertants per nanomole (lnR).

#### 3.2.2. Anticancer Activity

The endpoint considered here is IC50 which represents the concentration of the agent necessary to reduce cell viability by 50% against Murine P388 Leukemia (in vitro cytotoxic activity). The endpoint is expressed on a logarithmic scale (pIC50).

#### 3.2.3. Blood–Brain Barrier (BBB)

The database for BBB permeation (*n* = 291) is taken from the literature [105].

QSAR models for the above-listed endpoints are based on the following descriptor:(3)$$\begin{array}{l} DCW(T*,N*)=CW(BOND)+CW(NOSP)+CW(HALO)+CW(PAIR)+\\ {}^{}{}^{}{}^{}{}^{}{}^{}{}^{}{}^{}\sum CW(S_{k})+\sum CW(SS_{k}) \end{array}$$

Here, the global SMILES attributes are the following *BOND*, *NOSP*, *HALO*, and *PAIR*. The *S_k_* and *SS_k_* are local SMILES attributes. Table 4 and Table 5 contain comments on these attributes. The *CW(BOND), CW(NOSP), CW(HALO), CW(PAIR), CW(S_k_),* and *CW(SS_k_)* are correlation weights of the above-listed attributes.

The scheme of estimation of similarity and dissimilarity for the above-mentioned endpoints demonstrated by Table 6 is adapted from [105].

Table 7 contains numerical measures of similarity and dissimilarity of the corresponding endpoints (Table 6).

The similarity of endpoints defined according to suggested scheme can become the beginning of “a next generation” of the QSPR/QSAR evolution.

### 3.3. Gender-Oriented QSAR Models

Usually, the categorization of eco-toxic effects is related to a different animal (fishes, birds, and insects). However, in addition, at least for animals, categorization related to sex also may be useful from practical and theoretical points of view. QSAR models of carcinogenicity separately for male and female rats can have wide applications for both the agriculture and theoretical biochemistry [69]. The matrix to recognize the difference of corresponding models build up by scheme analogous to the above-mentioned scheme applied for Table 6. The promoters of the increase of pTA50 have been examined separately in the cases of male and female rats. In both cases, symbol ‘1’ means the stable positive correlation weight, whereas symbol ‘0’ means the stable negative correlation weight. Table 8 contains the results of the above-mentioned computational experiments.

Examples of molecular features acted in a different manner for male rats and female rats are the following (i) BOND1010000; (ii) HALO00000000; (iii) NNC-C…303.; and (iv) NNC-C…321. This information can be useful, e.g., for developers of corresponding biocides. Algorithms able to generate gender-oriented models may have wider applications (e.g., drug design).

### 3.4. The Simplicity or the Efficiency: Which is Better?

QSAR should be assessed as a surrogate of a real experiment. QSAR aimed to measure an endpoint value. However, to expect adequate prediction physicochemical and biochemical behavior of an arbitrary substance by means of the QSPR/QSAR-model is naive.

Despite the above-mentioned thesis, QSPR/QSAR has become an integral part of modern science as a tool to detect “fuzzy tendencies” in the behavior of groups of substances. This fact logically echoes the theory of fuzzy sets [108]. This is not surprising, as fuzzy set theory has success in solving some problems of QSPR/QSAR analysis [109,110,111].

One can extract two components in the total big variety of QSAR studies: (i) “extensive” studies and (ii) “intensive” studies. The aim of “extensive” studies is the integration of the results of applying current approaches to solve practical tasks. The aim of “intensive” studies is attempting to develop new conceptions of the QSPR/QSAR analysis. Naturally, a small part of the results of the “intensive” studies gradually become a tool of robust “extensive” studies.

Nowadays, multi-target QSAR is a part of “intensive” studies [96,97,98,99,100,101,102,103,104]. The development of criteria of the predictive potential of models (Table 3) also is a part of the “intensive” studies. Maybe, search for the similarity of endpoints [105,106,107], also, will become part of “intensive” QSPR/QSAR researches.

## 4. Conclusions

The evolution of the field of QSPR/QSAR has two components: intensive and extensive. The intensive component is responsible for developing the quality and epistemology potential of various QSPR/QSAR approaches. The multi-target QSAR is the perspective field of the evolution of the QSAR theory and practices. Other perspective components of the “intensive” evolution of the QSPR/QSAR are (i) applying fuzzy set theory; (ii) developing statistical methods to detect similarity of biochemical endpoints; and (iii) extending “input data” for QSPR/QSAR by means of taking into account experimental conditions and circumstances also can be a component of intensive evolution of the QSPR/QSAR.

## Figures and Tables

**Figure 1 molecules-25-01292-f001:**
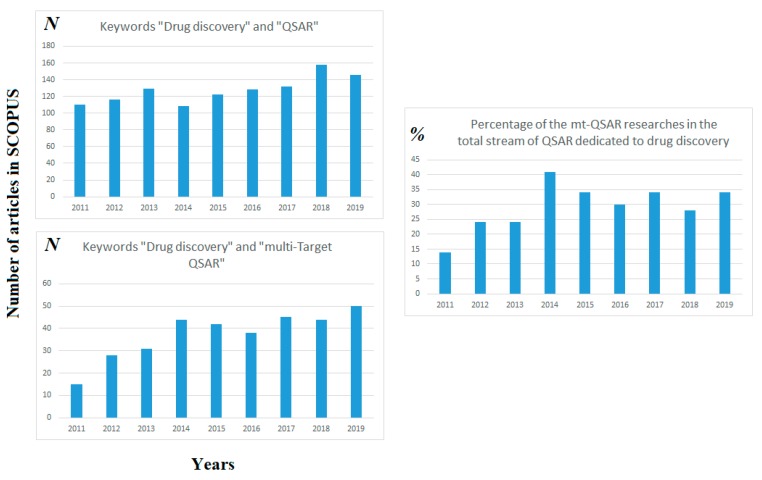
Comparison of frequencies of using quantitative structure–activity relationships (QSAR) and multi-target QSAR (mt-QSAR) in drug discovery researches.

**Table 1 molecules-25-01292-t001:** Distribution of 87 anticancer inhibitors [60] into training and validation sets.

**Split #1**	**Training set** = 1, 4, 6, 7, 8, 9, 12, 13, 14, 15, 16, 17, 18, 19, 20, 21, 23, 25, 27, 29, 30, 31, 33, 34, 36, 37, 38, 40, 41, 42, 45, 46, 48, 51, 52, 54, 55, 56, 57, 59, 60, 61, 63, 64, 65, 67, 69, 70, 73, 74, 75, 77, 90, 94, 98, 99, 109, 112, 116, 117, 118, 120, 121, 122, 123, 124, 126, 130, 136; **Validation set** = 2, 3, 5, 10, 11, 22, 26, 32, 35, 39, 43, 47, 68, 71, 92, 103, 125, 143
**Split #2**	**Training set** = 1, 6, 7, 8, 9, 12, 13, 14, 15, 16, 18, 19, 23, 25, 27, 31, 33, 34, 36, 40, 41, 42, 45, 46, 48, 51, 54, 55, 56, 57, 59, 61, 63, 65, 67, 69, 73, 74, 75, 77, 98, 109, 112, 116, 117, 121, 123, 124, 130, 136, 5, 10, 11, 22, 26, 32, 39, 43, 47, 68, 71, 92, 103, 125, 143; **Validation set** = 4, 17, 20, 21, 29, 30, 37, 38, 52, 60, 64, 70, 90, 94, 99, 118, 120, 122, 126, 2, 3, 35
**Split #3**	**Training set** = 1, 4, 6, 7, 8, 9, 12, 13, 14, 15, 16, 17, 18, 19, 21, 23, 25, 27, 29, 31, 36, 37, 38, 40, 41, 42, 45, 48, 51, 54, 56, 57, 59, 60, 64, 65, 69, 70, 73, 74, 75, 77, 94, 98, 99, 109, 116, 118, 121, 124, 130, 136, 2, 5, 10, 11, 26, 32, 35, 39, 43, 47, 68, 71, 125, 143; **Validation set** = 20, 30, 33, 34, 46, 52, 55, 61, 63, 67, 90, 112, 117, 120, 122, 123, 126, 3, 22, 92, 103

**Table 2 molecules-25-01292-t002:** The predictive potential of different approaches observed for different splits.

Method	Split	Number of Compounds in Validation Set	Determination Coefficient for Validation Set
**3D-QSAR [60]**	#1	18	0.77
**Method 1**	#1	18	0.43
**Method 2**	#1	18	0.53
**Method 1**	#2	22	0.84
**Method 1**	#3	21	0.81
**Method 2**	#2	22	0.82
**Method 2**	#3	21	0.85

**Table 3 molecules-25-01292-t003:** Statistical criteria of the predictive potential for the quantitative structure–property activity relationships (QSPR/QSAR) models.

Criterion of the Predictive Potential	Reference
$R=\frac{n\sum \;\mathrm{xy}-\sum x\sum y}{\sqrt{(n\sum x^{2}-{\left(\sum x\right)}^{2}(n\sum y^{2}-{\left(\sum y\right)}^{2}}}$	[75]
$CCC=\frac{2\sum \;\left(x-\bar{x}\right)\left(y-\bar{y}\right)}{\sum \left(x-\bar{x}\right){\;}^{2}+\sum \left(y-\bar{y}\right){\;}^{2}+n\left(\bar{x}-\bar{y}\right){\;}^{2}}$	[76]
$R_{0}^{2}=1-\frac{\sum {\left({\tilde{y}}_{i}-y_{i}^{ro}\right)}^{2}}{\sum {\left({\tilde{y}}_{i}-\bar{\tilde{y}}\right)}^{2}}$ ${R^{\prime }}_{0}^{2}=1-\frac{\sum {\left(y_{i}-{\tilde{y}}_{i}^{ro}\right)}^{2}}{\sum {\left(y_{i}-\bar{y_{i}}\right)}^{2}}$ $k=\frac{y_{i\;}\tilde{y_{i}}}{{\tilde{y}}_{i}^{2}}$ $k^{\prime }=\frac{y_{i\;}\tilde{y_{i}}}{y_{i}^{2}}$	[77]
$Q^{2}=1-\frac{\sum (y_{k}-{ý}_{k}){\;}^{2}}{\sum (y_{k}-{\bar{y}}_{k}^{\;}){\;}^{2}}$ $Q_{F1}^{2}=1-\frac{\left[\sum_{i=1}^{N_{EXT}}\left({\overset{´}{y}}_{i}-y_{i}\right){\;}^{2}\right]/N_{EXT}}{\left[\sum_{i=1}^{N_{EXT}}\left(y_{i}-\bar{y}{\;}_{TR}\right){\;}^{2}\right]/N_{EXT}}$ $Q_{F2}^{2}=1-\frac{\left[\sum_{i=1}^{N_{EXT}}\left({\overset{´}{y}}_{i}-y_{i}\right){\;}^{2}\right]/N_{EXT}}{\left[\sum_{i=1}^{N_{EXT}}\left(y_{i}-\bar{y}{\;}_{EXT}\right){\;}^{2}\right]/N_{EXT}}$ $Q_{F3}^{2}=1-\frac{\left[\sum_{i=1}^{N_{EXT}}\left({\overset{´}{y}}_{i}-y_{i}\right){\;}^{2}\right]/N_{EXT}}{\left[\sum_{i=1}^{N_{TR}}\left(y_{i}-\bar{y}{\;}_{TR}\right){\;}^{2}\right]/N_{TR}}$	[78]
$r_{m}^{2}=r^{2}\left(\;1-\sqrt{\left|r^{2}-r_{0}^{2}\right|\;}\right)$ $\bar{R_{m}^{2}}=\frac{R_{m}^{2}\left(x,y\right)-R_{m}^{2}\left(y,x\right)}{2}$ $\Delta R_{m}^{2}=\left|R_{m}^{2}\left(x,y\right)-R_{m}^{2}\left(y,x\right)\right|$	[79]
IICCLB=rCLBmin(M −AE CLB,M +AE CLB)max(M −AE CLB,M +AE CLB) M −AE CLB=1N −∑k=1N −| Δk|,Δk0;N − is the number of Δk<0 M +AE CLB=1N +∑k=1N +| Δk|, Δk0;N + is the number of Δk≥0 $\Delta_{k}=observed_{k}-calculated_{k}$	[80]

**Table 4 molecules-25-01292-t004:** Simplified molecular input-line entry system (SMILES) attributes applied to build up a model.

SMILES Attribute	Comments
*S_k_*	One symbol or two symbols which cannot be examined separately in SMILES, e.g., Cl, Br, etc.
*SS_k_*	A combination of two connected *S_k_*
BOND	Descriptor reflects the presence in SMILES of the following symbols: ‘@’, ‘=’, and ‘#’ (i.e. presence of different bonds)
NOSP	Descriptor reflects the presence of the following chemical elements nitrogen (i.e., symbol ‘N’), oxygen (i.e., symbols ‘O’), Sulfur (i.e., symbol ‘S’), and phosphorus (i.e., symbol ‘P’)
HALO	Descriptor reflects the presence of fluorine (i.e., symbol ‘F’), chlorine (i.e., symbols ‘Cl’), bromine (i.e., symbols ‘Br’), and iodine (i.e., ‘I’)
PAIR	Descriptor reflects simultaneous the presence of pair of the above elements (i.e. details related to BOND, NOSP, and HALO, without any details about their places in molecular structure)

**Table 5 molecules-25-01292-t005:** Generalized representation of above SMILES attributes for Clc1cc(Cl)ccc1C(O)=O.

ID	Attribute	1	2	3	4	5	6	7	8	9	10	11	12
**1**	*S_k_*	C	l	.	.	.	.	.	.	.	.	.	.
		c	.	.	.	.	.	.	.	.	.	.	.
		1	.	.	.	.	.	.	.	.	.	.	.
		c	.	.	.	.	.	.	.	.	.	.	.
		c	.	.	.	.	.	.	.	.	.	.	.
		(	.	.	.	.	.	.	.	.	.	.	.
		C	l	.	.	.	.	.	.	.	.	.	.
		(^*^	.	.	.	.	.	.	.	.	.	.	.
		c	.	.	.	.	.	.	.	.	.	.	.
		c	.	.	.	.	.	.	.	.	.	.	.
		c	.	.	.	.	.	.	.	.	.	.	.
		1	.	.	.	.	.	.	.	.	.	.	.
		C	.	.	.	.	.	.	.	.	.	.	.
		(	.	.	.	.	.	.	.	.	.	.	.
		O	.	.	.	.	.	.	.	.	.	.	.
		(	.	.	.	.	.	.	.	.	.	.	.
		=	.	.	.	.	.	.	.	.	.	.	.
		O	.	.	.	.	.	.	.	.	.	.	.
**2**	*SS_k_*	c	.	.	.	C	l	.	.	.	.	.	.
		c	.	.	.	1	.	.	.	.	.	.	.
		c	.	.	.	1	.	.	.	.	.	.	.
		c	.	.	.	c	.	.	.	.	.	.	.
		c	.	.	.	(	.	.	.	.	.	.	.
		C	l	.	.	(	.	.	.	.	.	.	.
		C	l	.	.	(	.	.	.	.	.	.	.
		c	.	.	.	(	.	.	.	.	.	.	.
		c	.	.	.	c	.	.	.	.	.	.	.
		c	.	.	.	c	.	.	.	.	.	.	.
		c	.	.	.	1	.	.	.	.	.	.	.
		c	.	.	.	1	.	.	.	.	.	.	.
		C	.	.	.	1	.	.	.	.	.	.	.
		O	.	.	.	(	.	.	.	.	.	.	.
		O	.	.	.	(	.	.	.	.	.	.	.
		=	.	.	.	(	.	.	.	.	.	.	.
		=	.	.	.	=	.	.	.	.	.	.	.
**3**	*BOND*	B	O	N	D	1	0	0	0	0	0	0	0
**4**	*NOSP*	N	O	S	P	0	1	0	0	0	0	0	0
**5**	*HALO*	H	A	L	O	0	1	0	0	0	0	0	0
**6**	*PAIR*	+	+	+	+	C	l	.	.	O	=	=	=
		+	+	+	+	C	l	.	.	B	2	=	=
		+	+	+	+	O	.	.	.	B	2	=	=

^*)^ Used only “(“, not ‘)’; ^**)^ Symbols in *SS_k_* are placed according to ASCII code, in order to avoid situation wrong interpretations AB and BA as non-equivalent features.

**Table 6 molecules-25-01292-t006:** Definition of similarities to models for mutagenicity, anticancer activity and blood–brain barrier (BBB). Here, model-1, denoted m1; model-2, denoted m2. The “m1.1” means first run of optimization for endpoint 1. Each plus denotes a promoter of an increase for endpoints (#1 or #2). Each minus denotes a promoter for a decrease for endpoints (#1 or #2).

	Attributes, SA_k_	m1.1	m1.2	m1.3	m2.1	m2.2	m2.3
Mutagenicity (#1) vs. Anticancer Activity (#2)							
1	1...........	+	+	+	+	+	+
2	c...2.......	+	+	+	+	+	+
3	c...(.......	+	+	+	+	+	+
4	3...........	+	+	+	+	+	+
5	C...........	+	+	+	+	+	+
6	1...(.......	+	+	+	+	+	+
7	C...1.......	+	+	+	+	+	+
8	C...3.......	+	+	+	+	+	+
9	Cl..(.......	+	+	+	+	+	+
10	Cl..........	+	+	+	+	+	+
1	c...........	+	+	+	−	−	−
2	O...........	+	+	+	−	−	−
3	O...(.......	+	+	+	−	−	−
4	N...(.......	−	−	−	+	+	+
5	++++N---O===	−	−	−	+	+	+
6	NOSP11000000	−	−	−	+	+	+
7	C...(.......	−	−	−	+	+	+
8	C...C.......	−	−	−	+	+	+
Mutagenicity (#1) vs. BBB (#2)							
1	1...........	+	+	+	+	+	+
2	BOND00000000	+	+	+	+	+	+
3	HALO00000000	+	+	+	+	+	+
4	NOSP10000000	+	+	+	+	+	+
5	1...(.......	+	+	+	+	+	+
6	++++CL--N===	+	+	+	+	+	+
7	-...........	+	+	+	+	+	+
8	=...(.......	+	+	+	+	+	+
9	C...1.......	+	+	+	+	+	+
10	BOND10000000	+	+	+	+	+	+
11	Cl..(.......	+	+	+	+	+	+
12	Cl..........	+	+	+	+	+	+
13	N...+.......	+	+	+	+	+	+
14	N...........	−	−	−	−	−	−
1	O...........	+	+	+	−	−	−
2	O...(.......	+	+	+	−	−	−
3	N...1.......	+	+	+	−	−	−
4	[...+.......	+	+	+	−	−	−
5	NOSP11000000	−	−	−	+	+	+
6	C...(.......	−	−	−	+	+	+
7	C...C.......	−	−	−	+	+	+
BBB (#1) vs. anticancer activity (#2)							
1	C...C.......	+	+	+	+	+	+
2	C...(.......	+	+	+	+	+	+
3	1...........	+	+	+	+	+	+
4	C...1.......	+	+	+	+	+	+
5	C...=.......	+	+	+	+	+	+
6	++++N---B2==	+	+	+	+	+	+
7	C...2.......	+	+	+	+	+	+
8	NOSP11000000	+	+	+	+	+	+
9	1...(.......	+	+	+	+	+	+
10	O...C.......	+	+	+	+	+	+
11	2...(.......	+	+	+	+	+	+
12	4...........	+	+	+	+	+	+
13	Cl..........	+	+	+	+	+	+
14	Cl..(.......	+	+	+	+	+	+
15	++++S---B2==	+	+	+	+	+	+
16	HALO01000000	+	+	+	+	+	+
17	++++F---B2==	+	+	+	+	+	+
18	++++F---N===	+	+	+	+	+	+
19	HALO10000000	+	+	+	+	+	+
20	N...4.......	+	+	+	+	+	+
21	++++CL--S===	+	+	+	+	+	+
22	(...........	−	−	−	−	−	−
23	O...........	−	−	−	−	−	−
24	O...(.......	−	−	−	−	−	−
25	5...........	−	−	−	−	−	−
26	C...5.......	−	−	−	−	−	−
1	++++Cl--B2==	+	+	+	−	−	−
2	F...(.......	+	+	+	−	−	−
3	++++F---Cl==	+	+	+	−	−	−
4	++++O---B2==	−	−	−	+	+	+
5	2...........	−	−	−	+	+	+
6	=...2.......	−	−	−	+	+	+
7	3...(.......	−	−	−	+	+	+
8	++++O---S===	−	−	−	+	+	+

**Table 7 molecules-25-01292-t007:** The matrix of similarity for examining endpoints.

Similarity			
	Mutagenicity	Anticancer Activity	Blood–Brain Barrier
**Mutagenicity**	41	10	14
**Anticancer activity**	10	61	26
**Blood–brain barrier**	14	26	92
**Dissimilarity**			
**Mutagenicity**	11	8	7
**Anticancer activity**	8	24	8
**Blood–brain barrier**	7	8	52

**Table 8 molecules-25-01292-t008:** Promoters for increase carcinogenicity in male rats (MR) and female rats (FR).

Promoters of Carcinogenicity Increase	Male Rats, MR									Total MR	Female Rats, FR									Total FR
	Split 1			Split 2			Split 3				Split 1			Split 2			Split 3			
	1	2	3	1	2	3	1	2	3		1	2	3	1	2	3	1	2	3	
Molecular features extracted from SMILES																				
1...(.......	0	0	0	0	0	0	0	0	0	0	1	1	1	1	0	1	1	0	1	7
2...(.......	1	1	1	0	0	0	1	1	1	6	0	0	0	0	0	0	0	0	0	0
2...1.......	1	1	0	0	0	1	1	1	0	5	0	0	0	0	0	0	0	0	0	0
C...1.......	0	0	0	0	0	0	0	0	0	0	1	1	0	0	0	0	0	0	0	2
C...2.......	0	1	0	1	1	0	0	1	0	4	0	0	0	0	0	0	0	0	0	0
N...=.......	1	0	0	0	1	1	1	0	0	4	0	0	0	0	0	0	0	0	0	0
N...1.......	0	0	0	1	1	1	0	0	0	3	0	0	0	0	0	0	0	0	0	0
HALO00000000	1	1	1	1	1	1	1	1	1	9	0	0	0	0	0	0	0	0	0	0
BOND00000000	1	0	1	0	0	0	1	0	1	4	1	1	1	1	1	1	1	1	1	9
BOND10000000	0	0	0	0	0	0	0	0	0	0	1	1	0	1	1	1	0	0	0	5
BOND10100000	1	1	1	1	1	1	1	1	1	9	0	0	0	0	0	0	0	0	0	0
Molecular features (invariants) extracted from molecular graph*																				
C5......0...	0	0	0	1	0	1	0	0	0	2	1	0	0	1	1	1	0	0	0	4
C6......0...	1	1	1	1	1	1	1	1	1	9	1	1	1	1	1	1	1	1	0	8
NNC-C...101.	1	1	1	1	1	1	1	1	1	9	0	0	0	0	0	0	1	1	0	2
NNC-C...110.	0	0	0	0	0	0	0	0	0	0	0	0	0	1	1	1	1	0	1	5
NNC-C...211.	1	1	1	1	1	1	1	1	1	9	1	1	1	1	1	1	1	1	1	9
NNC-C...303.	1	1	1	1	0	0	1	1	1	7	0	0	0	0	0	0	0	0	0	0
NNC-C...321	0	0	0	0	0	0	0	0	0	0	0	1	1	1	1	1	1	1	0	7
NNC-O...101	0	0	0	0	0	0	0	0	0	0	0	1	1	0	0	0	0	0	0	2
Summation										63										50

*) Detailed description for C5……… and C6……… represented in [74]; detailed description for NNC-Y…xxx represented in [80].

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
