# Peer review of "QSPR/QSAR: State-of-Art, Weirdness, the Future"

_molecules, 2020, doi:10.3390/molecules25061292_

Round 1

Reviewer 1 Report

The manuscript „QSPR/QSAR: state-of-art, weirdness, the future” by Andrey A. Toropov and Alla P. Toropova has large potential to become an interesting, inspiring mini-review of the current state of the QSPR/QSAR field. The authors do not hesitate to point out weak points of the current QSPR techniques, and to propose possible solutions based on their own research. However, the manuscript is not yet suitable for publication in its current form. I suggest preparing a major revision of the manuscript according to the suggestions below, and then the manuscript can become a valuable position in the “Molecules” journal.

Scientific and methodological issues:

1. The very important section 2 begins with stating three “main unpleasant peculiarities of QSPR/QSAR analysis”. A moment below, three points of “weirdness” are introduced and discussed in Sections 2.1, 2.2 and 2.3. There is no clear connection between these two “trinities” of peculiarity and weirdness. Please include a short paragraph defining how a given peculiarity (e.g. possibility of overtraining) is influencing the points of “weirdness” (maybe more and more predictive potential measures make it easy to overlook the overtraining, etc.?)

2. Moreover, the peculiarities and weirdnesses are no more discussed in Section 3 “Discussion”. It is necessary to show how the research reviewed in Section 3 aims to address the issues raised in Section 2.

3. Much of the space of Section 3 is taken by Tables 5 and 6. If I understand correctly, the data are taken from Ref. 105. Is it possible to move them to the Supporting Information for the interested readers, or even delete them and make just the note that they are taken from Ref. 105?

Presentation issues:

4. The first paragraph of the Introduction (lines 20-26) seems to be of a very general philosophical nature, and should be either rewritten to include direct connection to the main topic, or replaced by a more suitable, more specific introduction.

5. The manuscript has to be language-checked. I am not a native English speaker, so I do not dare to prepare a full list of items to be corrected, but in my opinion there are many of them. Many are of the type of “wrong use of singular / plural” or “passive vs. active”, for example: line 68-69 “Attempts… are performing”; line 73: “The distribution have key influence”. Sometimes a word is missing, e.g. line 77 “According many authors” (according to); line 117 “majority of QSPR/QSAR have built up” (have been built up).

End of the reviewer remarks.

Author Response

To Reviewer #1

Dear Madam or Sir,

Thank you very much for deep analysis of our work.

We have attempted to revise our manuscript according to your remarks.

Our responses placed along your text.

All modifications are indicated by red in the revised manuscript.

Sincerely yours,

Alla P. Toropova on behalf of the authors

1.

The manuscript „QSPR/QSAR: state-of-art, weirdness, the future” by Andrey A. Toropov and Alla P. Toropova has large potential to become an interesting, inspiring mini-review of the current state of the QSPR/QSAR field. The authors do not hesitate to point out weak points of the current QSPR techniques, and to propose possible solutions based on their own research. However, the manuscript is not yet suitable for publication in its current form. I suggest preparing a major revision of the manuscript according to the suggestions below, and then the manuscript can become a valuable position in the “Molecules” journal.

Scientific and methodological issues:

  1. The very important section 2 begins with stating three “main unpleasant peculiarities of QSPR/QSAR analysis”. A moment below, three points of “weirdness” are introduced and discussed in Sections 2.1, 2.2 and 2.3. There is no clear connection between these two “trinities” of peculiarity and weirdness. Please include a short paragraph defining how a given peculiarity (e.g. possibility of overtraining) is influencing the points of “weirdness” (maybe more and more predictive potential measures make it easy to overlook the overtraining, etc.?)

Response 1:

 We have added the following fragments of text:

Examination of several splits decrease probability of “chance correlations”: solely one good correlation easily can become chance correlation, however, three (five, six, seven...) good correlations hardly can be "chance correlations".

In other words, contradictions in the recommendations of various criteria force the researcher to search for truth (i.e. the best choice) in a greater maze of possibilities.

It is to be noted, however, in some cases the molecular structure is not informative to build up predictive model of endpoints [81-95].  Meantime, the definition of a model as a mathematical function of experimental conditions (which can be optimized  by experimentalist) are more short and consequently more attractive way to solve corresponding task.

  1. Moreover, the peculiarities and weirdnesses are no more discussed in Section 3 “Discussion”. It is necessary to show how the research reviewed in Section 3 aims to address the issues raised in Section 2.

Response 2:

We have added to Discussion (at the beginning) the following paragraph:

 The above-mentioned unpleasant peculiarities and weirdness are interacted. To avoid unpleasant peculiarities, one should build up a model  without the above weirdnesses, namely, (i) one should study several different splits (into training and validation sets); (ii) one should select a group of criteria of predictive potential which are agree each other; and (iii) one should take into account all conditions which impact corresponding endpoint (not only molecular structure). However, these actions are not enough to solve all problems.

  1. Much of the space of Section 3 is taken by Tables 5 and 6. If I understand correctly, the data are taken from Ref. 105. Is it possible to move them to the Supporting Information for the interested readers, or even delete them and make just the note that they are taken from Ref. 105?

Response 3:

Unfortunately, it is not easy to understand these results without of Tables 5 and 6. It will not be friendly in relation to the reader, the discussion of these results without Tables 5 and 6.

Presentation issues:

  1. The first paragraph of the Introduction (lines 20-26) seems to be of a very general philosophical nature, and should be either rewritten to include direct connection to the main topic, or replaced by a more suitable, more specific introduction.

Response 4:

 We have rephrase the paragraph as the following:

Each science meets with internal and external contradictions. Correlations many times have served as a key to interpretation various phenomena. Expansion of information available for analysis (e.g. search space along “traditional" substances has been extended by nanomaterials) lead to the following question: whether correlation is useful or it will better to try to define a causality? [1]. Apparent, an answer this question doom to be non-completed.  

  1. The manuscript has to be language-checked. I am not a native English speaker, so I do not dare to prepare a full list of items to be corrected, but in my opinion there are many of them. Many are of the type of “wrong use of singular / plural” or “passive vs. active”, for example: line 68-69 “Attempts… are performing”; line 73: “The distribution have key influence”. Sometimes a word is missing, e.g. line 77 “According many authors” (according to); line 117 “majority of QSPR/QSAR have built up” (have been built up).

Response 5:

We have modified our text according to your remarks:

line 68-69:

“Attempts to solve problems related to the above "unpleasant peculiarities" of QSPR/QSAR are performing”

We have modified the sentence as the following:

“Attempts to solve problems related to the above "unpleasant peculiarities" of QSPR/QSAR have been performed. However, these attempts gave three weirdness points.“

line 73: “The distribution have key influence”.

We have corrected:

“The distribution has key influence for the statistical quality of QSPR/QSAR models”

line 77 “According many authors”

We have corrected: “According to many authors”

line 117 “majority of QSPR/QSAR have built up”

We have corrected: “majority of QSPR/QSAR has built up”

Reviewer 2 Report

It is courtesy to honor Hansch and Fujita, who created the first QSAR model developed by Hansch and Fujita (10.1021/ja01062a035)

One of the major problems with this manuscript is,

The mutagenicity study was based on TA98+S9 data and it has been observed that log TA98 depends on energy terms (10.1002/em.2850190107). However, this study did not take into account the energy factor.

Some phrases are plagiarized. E.g.

Line 151: The mutagenic activity in Salmonella typhimurium TA98+S9 microsomal reparation is expressed

This is the exact statement used in reference 105

A number of minor issues are,

Table 3 should list

  1. Cross-validation correlation coefficient (qcv2)
  2. Δrm2

References should be added to the paragraph (Lines 27-32)

Line 34: as well as many others conditions and circumstances

Please elaborate on that and add references

Line 131: Apparently, this conception has attractive advantages;

Please elaborate on that

Figure 1. Another graph that shows the ratio of multi-target QSAR/QSAR should be added.

Line 190

3.3. Gender-oriented QSAR models

If gender is a factor in developing QSAR models, other factors such as disease state and age should be taken into account as well.

Professional editing service can substantial increase the quality of this manuscript. E.g. some grammatical errors

Line 61: There are improvement of

Line 65: to get a group of descriptor

Line 143: can be expressed vie correlation weights

Line 213: for arbitrary substance by means of QSPR/QSAR is naively

Author Response

To Reviewer #2

Dear Madam or Sir,

Thank you very much for deep analysis of our work.

We have attempted to revise our manuscript according to your remarks.

Our responses placed along your text.

All modifications are indicated by red in the revised manuscript.

Sincerely yours,

Alla P. Toropova on behalf of the authors

2.

Comments and Suggestions for Authors

It is courtesy to honor Hansch and Fujita, who created the first QSAR model developed by Hansch and Fujita (10.1021/ja01062a035)

Response 1:

We have mentioned the work in first paragraph of Introduction. The corresponding reference is #10.

One of the major problems with this manuscript is,

The mutagenicity study was based on TA98+S9 data and it has been observed that log TA98 depends on energy terms (10.1002/em.2850190107). However, this study did not take into account the energy factor.

Response 2:

 We have used data that is available in the literature. We have no possibility to carry out corresponding experiments.

Some phrases are plagiarized. E.g.

Line 151: The mutagenic activity in Salmonella typhimurium TA98+S9 microsomal reparation is expressed

This is the exact statement used in reference 105

Response 3:

 We have rephrased the sentence:

The endpoint for QSAR analysis is the mutagenic potential. The mutagenic potential in Salmonella typhimurium TA98+S9 microsomal reparation represents by the natural logarithm of R, where R is the number of revertants per nanomole (lnR).

A number of minor issues are,

Table 3 should list

  1. Cross-validation correlation coefficient (qcv2)
  2. Δrm2

Response 4:

We have added corresponding formulae for these criteria In Table 3

References should be added to the paragraph (Lines 27-32)

Response 5:

We have added references in this paragraph:

 The quantitative structure – property activity relationships (QSPRs/QSARs) are relatively new field of natural sciences. There is a large group of aims associated with QSPRs/QSARs technique; main of these are probably the follows (i) prediction of physicochemical behavior of various substances in industry and their further ecologic impacts [2-9]; (ii) biochemical behavior of various substances in ecological and medicinal aspects [10]; (iii) selection of substances, which can be perspective candidates to defined roles [11,12].

Line 34: as well as many others conditions and circumstances

Please elaborate on that and add references

Response 6:

We have modified the paragraph as the following:

Results of traditional experiments were depended to properties of substances, masses, radiation; heat capacity, electronic, physicochemical and biochemical conditions as well as porosity, Zeta potential of nanomaterials, time of exposure, irradiation, darkness, etc. Computational experiments related to QSPR/QSAR concerned to “information conditions” (available datasets) and “statistical conditions” (diversity of substances in datasets), as well as preference of the user. 

Line 131: Apparently, this conception has attractive advantages;

Please elaborate on that

Response 7:

We have modified the paragraph as the following:

 Apparently, this conception has attractive advantages, since provide a user by extend list of information (i.e. expected numerical data for groups of endpoints which affect phenomenon under consideration, e.g. therapeutic effect, inhibition, biocide potential, etc.). Nonetheless, traditional approaches serve as basis to solve the task of building up multi-target QSARs, e.g. using multiple regression [99], partial least squares (PLS) [100], artificial neural networks (ANN) [101-103], and random forest [104].

Figure 1. Another graph that shows the ratio of multi-target QSAR/QSAR should be added.

Response 8:

We have added picture where percentage of multi-target QSAR/QSAR is indicated.

Line 190

3.3. Gender-oriented QSAR models

If gender is a factor in developing QSAR models, other factors such as disease state and age should be taken into account as well.

Response 9:

This is good idea. However, we had no corresponding data during preparation of the manuscript.

Professional editing service can substantial increase the quality of this manuscript. E.g. some grammatical errors.

Response 10:

We have corrected.

Line 61: “There are improvement of” has been changed to “There is an improvement”

Line 65: “to get a group of descriptor” has been changed to “to get a group of descriptors”

Line 143: “can be expressed vie correlation weights” has been changed to “can be expressed via correlation weights”

Line 213: “for arbitrary substance by means of QSPR/QSAR is naively” has been changed to “for an arbitrary substance by means of QSPR/QSAR-model is naively”

Reviewer 3 Report

The authors face some interesting problems, but their approach is mainly superficial, or philosophical/intuitive. It seems that the authors have grouped some features or examples they had in mind for some time. And then they comment them, but no practical results are provided and, as said, the topics are treated superficially.

Author Response

To Reviewer #3

Dear Madam or Sir,

Thank you very much for consideration of our work.

We have informed of the reader about some problems related to QSPR/QSAR. Having information about these problems some people can try to solve corresponding tasks. If this happens the manuscript is useful, is not?

Sincerely yours,

Alla P. Toropova on behalf of the authors

3.

Comments and Suggestions for Authors

The authors face some interesting problems, but their approach is mainly superficial, or philosophical/intuitive.

Response 1:

We have tried to use style able to be interesting not only for person who are researcher in the field of QSPR/QSAR but also arbitrary reader (e.g. a student or an experimentalist). 

It seems that the authors have grouped some features or examples they had in mind for some time. And then they comment them, but no practical results are provided and, as said, the topics are treated superficially.

Response 2:

We have prepared mini-review, not research article. We have discussed points which still have no unambigous responses and assessments, at present.

Reviewer 4 Report

The authors have reviewed on quantitative structure – property / activity relationships.

The problem mentioned in the draft is not very new. The authors have looked at, and suggest on the method as a machine learning /regression approach, but it should be something beyond that. The importance of understanding the real problem should never be undermined.

Author Response

To Reviewer #4

Dear Madam or Sir,

Thank you very much for consideration of our work.

We have informed of the reader about some problems related to QSPR/QSAR. Having information about these problems some people can try to solve corresponding tasks. If this happens the manuscript is useful, is not?

Sincerely yours,

Alla P. Toropova on behalf of the authors

4.

Comments and Suggestions for Authors

The authors have reviewed on quantitative structure – property / activity relationships.

The problem mentioned in the draft is not very new. The authors have looked at, and suggest on the method as a machine learning /regression approach, but it should be something beyond that. The importance of understanding the real problem should never be undermined.

Response:

We have discussed several problems related to QSPR/QSAR. Indeed, these problems are not extremely novel. However, these problems are important. QSPR/QSAR is an attractive tool to solve (at least partially) practical tasks. We deem possible ways to improve the above-mentioned tool can be interesting to many people. Factually, we have declare practical suitability of applying for all QSPR/QSAR approaches.

Round 2

Reviewer 1 Report

The revised version of the manuscript "QSPR/QSAR: state-of-art, weirdness, the future" by A. Toropov and A. Toropova is an improved version of the original submission. The authors have included most of the modifications that the reviewers have requested. I can recommend the publication of the manuscript on the scientific basis. I have, however, one major remark. The authors, up to my knowledge and in their own words, have included only those corrections to the English language and style that the reviewers specifically pointed out. My review explicitly states that "The manuscript has to be language-checked. I am not a native English speaker, so I do not dare to prepare a full list of items to be corrected, but in my opinion there are many of them", and I gave only some examples of the linguistic errors. There are many more left, and still in some parts the manuscript is not easy to read. I urge the authors to make one more spell check, using the help of a proficient English speaker, or use the services that are (possibly) available through the MDPI.

Author Response

Dear Madam or Sir,

Thank you very much for the consideration of our work.

We have tried to edit the text of our manuscript discussing the work with an expert on the English.

All modifications are indicated by red.

Sincerely yours,

Alla P. Toropova on behalf of the authors

Reviewer 2 Report

Some problems remain unsolved.

Author Response

Dear Madam or Sir,

Thank you very much for the consideration of our work.

We have tried to edit the text of our manuscript discussing the work with an expert on the English. Also we have tried to reduce the text that is may be classified as self-plagiarism. We hope that some problems become solved.

All modifications are indicated by red.

Sincerely yours,

Alla P. Toropova on behalf of the authors

Reviewer 3 Report

The paper contents has slightly improved due to some additions, but it has not changed substantially. The authors claim that it constitutes a mini-review. It is up to the Editor to decide if the journal or this particular issue admits a mini-review like this one.

Author Response

Dear Madam or Sir,

Thank you very much for the consideration of our work.

Sincerely yours,

Alla P. Toropova on behalf of the authors